# Review of the Application of Graphene-Based Coatings as Anticorrosion Layers

**Karolina Ollik * and Marek Lieder** 

Department of Process Engineering and Chemical Technology, Faculty of Chemistry, Gdansk University of Technology, 11/12 Gabriela Narutowicza Street, 80-233 Gdansk, Poland; lieder@pg.edu.pl

*   Correspondence: karolina.ollik@pg.edu.pl

**Abstract:** Due to the excellent properties of graphene, including flexibility that allows it to adjust to the curvature of the substrate surface, chemical inertness, and impermeability, graphene is used as an anticorrosion layer. In this review, we present the current state-of-the-art in the application of graphene in the field of protective coatings. This review provides detailed discussions about the protective properties of graphene coatings deposited by different methods, graphene-based organic coatings, the modification of graphene-based coatings, and the effects of graphene functionalization on the corrosion resistance of protective coatings.

**Keywords:** corrosion; graphene coatings; modified graphene coatings; protective properties

---

## 1. Characterization of Graphene

Graphene is one of many allotropic varieties of carbon, considered as single-layer carbon atoms, arranged in a honeycomb lattice. It is a material that reveals outstanding properties, i.e., high mechanical strength of 1100 GPa, high thermal conductivity, high carrier mobility, high surface area, very dense network, and others. Due to these properties, graphene can be applied in different branches of industry, including electronics [1], deoiling high salinity emulsions [2], gas permeability [3], catalysis [4], electrodialysis [5], biomimetics [6], filtration [7], energy [8] and hydrogen [9] storage, biologic sensors [10], or as anticorrosion layer [11].

Graphene is characterized by several properties that make it a perfect protective layer. For example, surfaces coated with graphene can retain their optical appearance due to the high light transmission of graphene (97%) in a wide range of the electromagnetic spectrum. Further, graphene is flexible, allowing graphene coatings to conform to the roughness and curvature of the substrate surface. The delocalization of the electron cloud in graphene causes the aromatic C=C bond network to extend across the entire basal plane, endowing graphene with thermodynamic stability. This chemical inertness is necessary for its use as a protective coating. Studies have shown that graphene is more stable than diamond when exposed to high pressure and superheated water [12]. In addition to its chemical inertness, graphene is impermeable; the dense graphene lattice acts as a barrier, blocking even helium atoms, the smallest of atoms [13] (Figure 1).

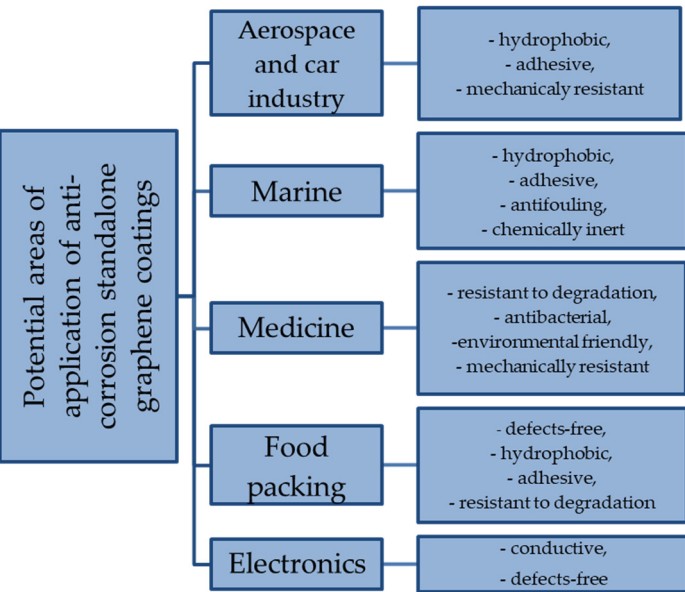

**Figure 1.** Potential areas of application of anticorrosion standalone graphene coatings.

Graphene can be synthesized by numerous methods including micromechanical cleavage, liquid-phase exfoliation, chemical vapor deposition (CVD), and chemical methods (Table 1).

In recent years, many research efforts have focused on the synthesis of graphene layers by a variety of methods. However, this review focuses on graphene coatings, graphene-based anticorrosion organic coatings, and strategies for their modification. The functionalization of standalone graphene coatings has not yet been reported. This work aims to (1) present the state-of-the-art in the area of protective graphene coatings and (2) identify the problems arising from the preparation of these coatings and their effects on the protective properties of the coatings.

Other reviews have reported on the preparation methods of graphene layers, the corrosion protection mechanisms of graphene layers, anticorrosion graphene layers, and graphene-based organic coatings and their modifications [27–30]. In addition to corrosion protection, graphene coatings have been applied as flame-retardant coatings, wear/scratch-resistant coatings, antifouling coatings, pollutant-adsorption coatings, and antiseptic coatings [31]. A review of the anticorrosion properties of graphene and graphene applications expected in the near future was also published [32].

**Table 1.** Synthetic methods of graphene.

| Method | Description of Method | Pros and Cons | Literature References |
|---|---|---|---|
| Micromechanical Cleavage | Using adhesive tape, a single layer of graphene is peeled from highly ordered pyrolytic graphene | **Pros:** uncomplicated process; **Cons:** nonuniform thickness of layer, very small-scale production | [14,15] |
| Liquid-Phase Exfoliation | Graphene layers are obtained by the exfoliation of graphite in solvent. Because graphene is hydrophobic, different additives (surfactants or polymer) or appropriate solvents are used to obtain a stable and uniform suspension of graphene | **Pros:** simple technique; **Cons:** environmental pollution, poor quality | [16] |
| Chemical Vapor Deposition | The graphene layer is formed by the decomposition of hydrocarbons under high temperature. The carbon sources are various hydrocarbons such as ethylene, benzene, acetylene, and methane | **Pros:** large size, good quality, and purity, small production scale; **Cons:** high temperature, expensive process, sophisticated equipment, toxic by-products | [15,17] |
| Reduction Methods | The deoxygenation of graphene oxide (GO) occurs as a result of thermal treatment or treatment with chemical reagents such as hydrazine, sodium borohydride, vitamin C, hydroiodic acid, sodium or potassium hydroxide solution, urea, thiourea, or hydroxylamine | **Pros:** simple technique, economical, large scale production; **Cons:** poor quality, harmful and toxic reagents | [18–26] |

## 2. Graphene as an Anticorrosion Coating

Graphene was first studied as an anticorrosion layer in 2011. Since then, many studies have reported protective graphene layers deposited by various methods, including solvent-based techniques (e.g., spin coating, spray coating, dip coating, drop casting, and layer-by-layer deposition), CVD, rapid thermal annealing, and electrophoretic deposition (EPD); (Tables 2 and 3).

### 2.1. Graphene-Based Coatings Deposited by CVD

Chen et al. demonstrated the ability of graphene coatings deposited by CVD to protect the surfaces of Cu and Cu/Ni alloys from oxidation under different conditions (in air at a given temperature or in hydrogen peroxide) [59]. In general, research has indicated that these coatings act as diffusion barriers for harmful ions, while they are inert on the activity of aggressive solutions. Based on scanning electron microscopy images, no visible changes were observed on the coated surface after exposure to an aggressive environment; oxidation was only observed at the grain boundaries or edges [59]. Potentiodynamic polarization measurements reported by Singh Raman indicated that coating graphene layers on a copper substrate reduced the corrosion density by two orders of magnitude, thereby decreasing the oxidation of Cu [33]. These results are inconsistent with those of Kirkland, who found that graphene coatings did not greatly enhance the anticorrosion properties [34]. Moreover, the corrosion potential, which reflects the susceptibility of the surface to corrosion, became more negative after coating with graphene, indicating that the substrate became more susceptible to the action of aggressive ions [34]. Other researchers have studied the effect of the mechanical transfer of graphene layers deposited by CVD on the anticorrosion properties. Graphene layers were transferred from a copper substrate to a nickel surface. The results indicated that a single layer of graphene did not remarkably decrease the corrosion rate. This finding was attributed to the numerous defects created during the transfer of the graphene layer. However, the presence of two or four layers of graphene markedly enhanced the protective properties [35].

### 2.2. Electrophoretically Deposited Graphene Coatings

Graphene oxide (GO) is known to possess numerous oxygen-containing functional groups (among other carboxyl groups) on the basal plane and edge. However, during electrophoretic deposition, these species are removed as a result of Kolbe reactions [60]. After reaching the anode, the negatively charged electrons of the carboxyl groups are lost as a result of the oxidation of carboxylate species. Carbon dioxide is evolved during this process, leaving unpaired electrons on the surface of GO. These unpaired electrons migrate toward other unoccupied electrons. The mechanism of this reaction is as follows:

$$2RCOO^- \rightarrow 2RCOO\bullet + 2e^- \text{ (oxidation of carboxylate)} \tag{1}$$

$$2RCOO\bullet \rightarrow 2R\bullet + 2CO_2 \text{ (decarboxylation)} \tag{2}$$

$$2R\bullet \rightarrow R - R \text{ (dimerization of radicals)} \tag{3}$$

However, such a recombination of radicals is only possible if it occurs between two overlapping layers of graphene and not within the same layer [60]. This behavior should lead to an enhancement in interlayer integrity.

**Table 2.** Overview of protective graphene coatings deposited by different techniques.

| Methods | Substrate | Coating Thickness | Corrosion Measurements | Literature References |
|---|---|---|---|---|
| Chemical Vapor Deposition | Copper, nickel | 1–2 layers, Single or few layers | Potentiodynamic polarization, electrochemical impedance spectroscopy (EIS), cyclic voltammetry | [33–35] |
| Rapid Thermal Annealing | Copper | 1 layer | Potentiodynamic polarization, EIS | [36] |
| Electrophoretic Deposition | Copper, carbon steel, aluminum, NdFeB (Neodymium–Iron-Boron alloy) | <10 nm, 12.4 μm (20 V), 25.4 μm (30 V), 40 nm (2 V), 400 nm (5 V), 1.5 μm (15 V), 3 μm, 1 μm (1 min), 2.5 μm (2 min) single layer or few layers | Potentiodynamic polarization, EIS, cyclic corrosion test, weight loss measurements | [37–47] |
| Electrodeposition | Mild steel, stainless steel, low-carbon steel | 40 nm, 2.3 μm, | Potentiodynamic polarization, EIS | [48–52] |
| Spin Coating | Titanium substrates | Few layers | Potentiodynamic polarization | [53] |
| Spray Coating | Carbon steel | 300 μm | Linear polarization resistance, EIS | [54] |
| Dip Coating | Aluminum, low alloy steel | Single-layer and multilayer | Potentiodynamic polarization, EIS | [55,56] |
| Drop Casting | Mild steel | Not available | EIS, Tafel polarization | [57] |
| Layer-by-Layer Deposition | Polyethylene terephthalate (PET) | 30–40 nm | Oxygen transmittance rate | [58] |

**Table 3.** Overview of numerous factors and their impact on the corrosion resistance.

| Methods | $I_D/I_G$ | Oxygen Content (%) | SEM Analysis | Corrosion Resistance | Literature References |
|---|---|---|---|---|---|
| Chemical Vapor Deposition | Not available | Not available | Fold, wrinkles, edges | Lower corrosion rate than nickel and copper surface | [34] |
| Rapid Thermal Annealing | Less than ~0.1 | Not available | No significant changes (or defects) | Significantly reduced corrosion current density ($I_{corr}$) | [36] |
| Electrophoretic Deposition | −1.09 (GO powder), 1.71 (GO/steel) | C/O: 0.76 (GO), 1.4 (rGO coating) | Nonuniform | $I_{corr}$ ($\mu A \cdot cm^{-2}$): <br> 1.183 × 10$^4$ (steel), 4140 (rGO/steel) | [37,45] |
| | −1.03 (GO powder), 1.009 (GO/Cu) | Not available | Uniform, thin, transparent coating | 15.375 (Cu), 12.44 GO/Cu | |
| Electrodeposition | Not available | Partial reduction of GO during electrodeposition | Rough surface with no distinguishable delamination | $I_{corr}$ ($\mu A \cdot cm^{-2}$): <br> 0.0912(SS), 0.0108 (GO), 0.00268 (GO–Ni(OH)$_2$) | [49–52] |
| | Not available | Not available | Compact and free crack morphology | 6.138 (Ni), 0.956 ÷ 4.845 (Ni–GO, depend on GO amounts) | |
| | 1.33 (GO powder) 0.58 (rGO–Zn) | Reduction of GO during electrodeposition | Thin and few layers | 1.5 (bare steel), 1.6 (rGO), 0.007 (rGO–Zn) | |
| | 0.95 (GO), 1.27 (Co–Ni–P/GO) | Not available | Rough morphology | 14.33 (Co–Ni–P), 3.05 (Co–Ni–P/GO) | |
| Spin Coating | 1.028 (GO), 1.027 (NaGO) | Not available | Lesser wrinkles and wavy features than GO coating | Corrosion potential ($E_{corr}$) (V): <br> −0.509 (titanium substrates), −0.290 (GO), −0.200 (NaGO) | [53] |
| Spray Coating | 1.33 ÷ 1.42 (alumina–GO composites, depend on GO content) | Not available | Smaller amounts of pores for alumina–GO coatings than alone alumina coating | $I_{corr}$ ($\mu A \cdot cm^{-2}$): <br> 47.306 (alumina coating) 8.463 ÷ 1.0 × 10$^{-5}$ (alumina–GO coatings, depend on GO amounts) | [54] |

**Table 3.** *Cont.*

| Methods | $I_D/I_G$ | Oxygen Content (%) | SEM Analysis | Corrosion Resistance | Literature References |
|---------|-----------|---------------------|--------------|----------------------|-----------------------|
| Dip Coating | 0.95 (GO powder), 1.15 (GO/Al) | Most functional oxygen groups were removed during dip procedure | Not available | $I_{corr}$ ($\mu$A·cm$^{-2}$): 10.316 (bare Al), 8.324 $\times$ 10$^{-3}$ (G-coating) | [55] |
| Drop Casting | Not available | Not available | Uniform morphology metal coating, uniform and nonuniform morphology metal-GO coating (depending on time deposition on top metal coating) | $I_{corr}$ ($\mu$A·cm$^{-2}$): 29.8 (SnZn), 23.7 $\div$ 6.66 (SnZn–GO–SnZn, depending on time deposition of top SnZn layer); 8.03 (ZnNi), 6.66 $\div$ 3.26 (ZnNi–GO–ZnNi, depending on time deposition of top ZnNi layer) | [57] |
| Layer-by-Layer Deposition | Not available | Not available | Not available | OTR (cc·m$^{-2}$·d$^{-1}$) 8.119 (bare PET), 8.229–0.05 (depending on GO-PEI layers) | [58] |

As mentioned above, during this process, negatively charged GO plates migrate toward the positively charged anode. However, the direction of migration can change by adding calcium cations to the solution, making this a cathodic EPD process [37,38]. In general, electrophoretically deposited GO coatings protect different metal substrates from corrosion [39–41]. However, the corrosion resistance depends on the morphology, adhesion, and wettability of the coating. The quality of the coatings is determined by the EPD parameters. Raza et al. studied the effects of two parameters (applied voltage and suspension concentration) on the corrosion rate. They found that a too-low voltage or suspension concentration led to a nonuniform coating with insufficient coverage, whereas an excessively high voltage or solution concentration resulted in more defects in the coating along with poor coating adhesion to the surface or a rougher surface, respectively [11]. Other researchers showed that an overly long deposition time resulted in coating fragility and cracking, resulting in penetration by aggressive ions [42,43].

As mentioned earlier, the coating hydrophobicity is also an important factor in the anticorrosion performance. The hydrophilic nature of GO significantly decreases its protective properties [44]. Numerous studies have shown that increasing the degree of reduction of GO decreases the coating wettability and thus enhances the corrosion resistance. Thermally treated GO coatings deposited by EPD (200 °C, 4 h) show a higher degree of deoxygenation than untreated GO coatings. The removal of oxygen-containing functional groups was confirmed by Fourier-transform infrared spectroscopy measurements. The spectrum of the thermally reduced GO sample shows only two peaks at 1050 cm$^{-1}$ (C–O) and 1555 cm$^{-1}$ (C=O) with reduced intensity compared to the untreated sample, indicating the successful thermal reduction and restoration of $sp^2$ bonds [46]. Double reduction increases the hydrophobicity, adhesion strength, and density of the coating, which is beneficial for the protective properties [46]. The chemical reduction of GO coatings using NaBH$_4$ increases the degree of deoxygenation and thus also enhances the protective properties [45]. Naghdi et al. annealed GO coatings in an oven at 50, 100, 150, and 200 °C for 5 h. As expected, the measured water contact angle increased with increasing annealing temperature. Thermal treatment can also enhance the adhesion of the coating to the substrate via the formation of chemical bonds between the surface of the substrate and GO [38].

Despite their protective properties, electrophoretically deposited GO coatings show lower corrosion resistance than coatings deposited by CVD. This is explained by the defects in the surfaces of electrophoretically deposited GO and the weaker adhesion to the substrate [47].

*2.3. Graphene Coatings Deposited by Other Methods*

Protective graphene coatings can also be produced by rapid thermal annealing. Single-layer acetone derived graphene coatings in sodium chloride solution were characterized by significantly improved corrosion resistance; the corrosion current density was reduced by 56 and 59 times compared to the as-received and mechanically polished copper surfaces, respectively. The passivation mechanism of this coating is based on the suppression of the cathodic reaction. Studies have shown that multilayer graphene coatings are more reliable than single-layer graphene coatings. Oxygen molecules and chloride ions can reach the substrate surface via the grain boundaries and defects in the monolayer, resulting in its partial oxidation [36].

Layer-by-layer deposition is another route for fabricating graphene coatings. When layers of GO and poly(ethylenimine) (PEI) are deposited on a polyethylene terephthalate (PET) surface, the barrier properties are determined by the oxygen transmittance rate, which depends on the number of graphene layers. The oxygen transmittance rate decreases as the number of graphene layers increases. The GO/PEI composite composed of five layers was found to be a good barrier with the ability to decrease the permeability of oxygen [58].

Graphene protective coatings can also be synthesized by dip coating. Liu et al. reported that graphene dip coatings reduced the corrosion current density by almost three times compared to bare Al [55]. Other studies have also demonstrated the protective properties of multilayer graphene

films deposited by this method. Multilayer graphene layers were prepared from two kinds of GO solution synthesized by different techniques. The first sample was aged for five days (GO1) to ensure the complete oxidation of GO, while the second sample was unaged (GO2). After the deposition of three-layer graphene coatings, the corrosion rate of the noncoated steel substrate (282 mpy) was reduced to 3.5 mpy (GO1) and 22.5 mpy (GO2). Enhancing the oxidation process during synthesis resulted in fewer defects and better coverage, thereby increasing the corrosion resistance. For both GO1 and GO2, adding additional layers further decreased the corrosion rate. Corrosion measurements for coupons coated with three-layer graphene in different electrolytes were also carried out. When the environment was water at pH = 6.0, 3.5 wt.% NaCl, or 1 M NaOH, the corrosion resistance was significantly enhanced by the graphene coating. In contrast, the protective properties of the coating were weak in 0.1 N HCl. This can likely be attributed to the highly corrosive nature of the chloride ions present in HCl [56].

Despite the good anticorrosion properties of graphene coatings, they have two drawbacks, which can shorten the lifetime of these films. The presence of minor cracks and scratches on the surface of the coatings can cause strong corrosion in the exposed area. This phenomenon is caused by graphene nobler than metals and the anodic area (metal substrate) is significantly smaller than the cathodic area (graphene coating). In general, obtaining a perfect graphene layer without any significant damages is difficult. However, if such a coating is produced, the occurrence of slight damage may lead to local corrosion. Therefore, it is an important mitigation of these disadvantages for the usage of graphene as an anticorrosion layer. For example, applying polymer with graphene may break the galvanic coupling between graphene and metal. Another method employs the addition of zinc to graphene what shifts their potential to more negative values [61].

### 2.4. Other Methods Used to Deposit GO Films

In this section, we present pulsed laser annealing as a method (PLA), which can be used for deposition for graphene films. High-resolution scanning electron microscope (HR-SEM) revealed the formation of uniform, high-quality rGO layers without any wrinkles [62]. Further, the laser treatment of the GO film leads to a reduction of graphene oxide [63,64]. Studies were carried out with different average laser power (from 7 to 50 mW). Raman analysis was used to characterize defects in the sample. The lowest concentration of structural defects was observed for samples with a 50 mW power laser [64].

As mentioned above, graphene coatings are produced by various techniques (equilibrium methods). However, synthesis by these methods leads to the formation of many defects in the structure of the coating, which causes the acceleration of corrosion initiation. The presented nonequilibrium method of graphene oxide fabrication can replace them. The advantages of this process like homogenous and compact surface and high quality and purity can substantially enhance corrosion resistance.

### 2.5. Mechanically Properties of Graphene Layers

The presence of defects in the produced graphene coatings, like voids and wrinkles significantly limits the mechanical properties. However, modification of graphene oxide with poly(vinyl alcohol) (PVA) can enhance the failure resistance. This phenomenon results from the formation of hydrogen bonding between PVA chains and the oxidized domain of GO. These interactions between them lead to the bridging of a crack in GO, resulting in an increase of toughness properties [65]. Further, the chemical composition of GO contributes also to the enhancement of mechanical properties. Namely, adhesion strength as well as interfacial shear can be improved by the water content control between the graphene oxide layer which results from the formation of strong hydrogen bonding between GO-water-GO [66].

## 3. Inorganic Functionalization of Graphene Coatings

In this section, we present methods for the modification of graphene coatings and the effects of such functionalization on the coating protective properties. One method for the inorganic modification

of graphene is the introduction of nitrogen into the carbon framework, most often in the form of pyrridinc-N, pyrrolic-N, and graphitic-N. The hydrophobicity of the doped coating and the intact, defect-free surface prevents penetration by aggressive ions and thus prevent metal oxidation [67]. Further, these coatings deposited on the copper substrate are characterized by better anticorrosion performance compared to nonfunctionalized GO coatings; however, they are not much better compared to rGO coatings. This behavior is attributed to the catalytic activity toward oxygen reduction reaction [68]. Ren et al. showed that the anticorrosion properties of nitrogen-doped graphene coatings produced on the copper result from the low conductivity and continuous structure of the coating. These properties hinder the formation of galvanic cells while also increasing corrosion resistance [69] (Figure 2).

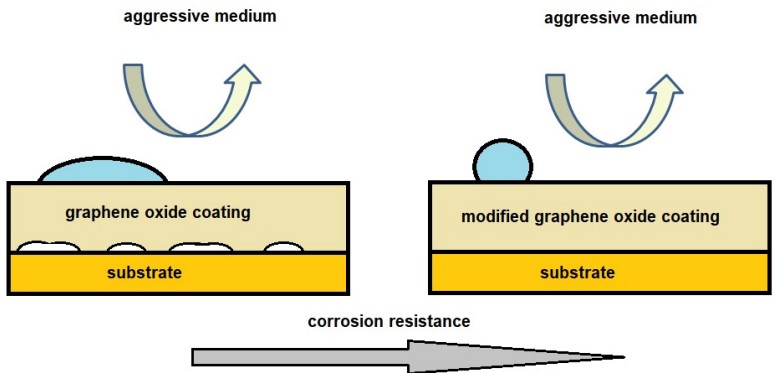

**Figure 2.** Mechanism protection of graphene coatings.

The protective properties of Ni/rGO–TiO$_2$ coatings on mild steel in organic acid were studied. Potentiodynamic polarization and EIS measurements showed enhanced corrosion resistance for these coatings compared to the Ni coating and neat substrate. The lowest corrosion rate was obtained for the Ni coating containing 20.4 wt.% RGO–TiO$_2$. The effects of the organic acid concentration, deposition current density, and temperature were also studied for the Ni coating with 20.4 wt.% RGO–TiO$_2$. The corrosion current density decreased with increasing citric acid concentration up to 0.06 M and then increased with further increases in concentration. At a low acid concentration, the reduced corrosion rate stems from the presence of a barrier layer and the adsorption of citric acid molecules on the electrode surface. The dependence of corrosion rate on acid concentration is completely different when the corrosive medium is acetic acid; in this case, the corrosion current density increases slightly as the acid concentration increases. This behavior contradicts the results published by Moussa [70], who found that the corrosion rate decreased as the acid strength increased. This is because, at a high concentration, acetic acid dissociates to release acetate ions, resulting in the passivation of the surface as a result of the adsorption of carboxylate molecules. As mentioned above, the effect of temperature was also studied. Increasing the temperature caused the corrosion rate to increase, independent of the corrosive medium. Under high temperatures, all reactions are accelerated. In measurements at different deposition current densities in both aggressive electrolytes, the lowest corrosion rate was obtained at 10 mA·cm$^{-2}$ [48]. Ni(OH)$_2$–GO and Ni–GO composites were also used as anticorrosion coatings and found to exhibit excellent protection efficiency; however, in the case of Ni(OH)$_2$–GO, the corrosion resistance is strictly related to the coating composition, which is affected by the parameters of pulse current deposition (i.e., pulse interval, time of deposition, and applied current). The deposition parameters have important influences on the concentrations of deposited species, the nucleation/deposition rate, and the accumulation of deposited particles. Thus, the coating properties, including thickness and the presence of defects, depend on the deposition conditions [49]. In the case of Ni–GO, the amount of added GO determines the corrosion rate. The GO in the hybrid coating enhances the grain growth of the layer along the low-energy (111) and (200) planes and

decreases grain growth along the relatively higher energy (220) plane. This makes the Ni–GO coatings less susceptible to penetration by aggressive electrolytes. High concentrations of added GO result in the agglomeration of GO flakes, leading to the formation of nonuniform coatings. In addition, after exceeding the optimum content of GO, galvanic coupling occurs because the area of the cathode (GO) is considerably larger than the area of the anode (nickel ions) [50]. Electrophoretically deposited GO coatings functionalized with magnesium ions were found to greatly reduce the corrosion rate (0.0116 mpy) compared to the blank Ti6Al4V substrate (0.0516 mpy). The presence of thick layers with surface functional groups (e.g., C=C and/or C=O) can effectively limit the access of corrosive media to the surface [71]. Titanium alloys coated with GO modified by sodium also exhibited protective properties [53]. The electrodeposition of reduced GO (rGO)/Zn nanocomposites on low-carbon steel reduced the corrosion rate from $0.035$ mm$\cdot$y$^{-1}$ for bare steel to $1.62 \times 10^{-4}$ mm$\cdot$y$^{-1}$. The excellent protective properties stem from the synergistic effect of reduced GO, which acts as a barrier, and the sacrificial dissolution of zinc as an anode [51].

The introduction of GO (0.5, 1, 1.5, or 2 wt.%) in alumina composites enhances the barrier properties. Corrosion protection was improved by nearly six orders of magnitude for the alumina composites containing 1.5 and 2.0 wt.% GO compared to the neat $Al_2O_3$ coating. This behavior results from the increase in hydrophobicity as the content of GO increases along with the thicker GO films formed as a result of the plasma spray process [54]. The codeposition of magnesium nitrate and GO leads to the formation of $Mg(OH)_2$/GO composite films, which show better anticorrosion performance than $Mg(OH)_2$ coatings. The more compact, intact structure of the $Mg(OH)_2$/GO film surface contributes to the enhanced barrier properties [72]. GO modified with zirconium oxide was used to enhance the corrosion resistance of zinc/aluminum coatings. The introduction of a hybrid provides a multilayer barrier that blocks aggressive molecules. Moreover, the use of KH560 [γ-(2,3-epoxy propylene oxide)propyltrimethoxysilane] to modify $ZrO_2$ + GO composite facilitates its dispersion in Zn/Al coating. The bonds formed between the silanol groups and metal can improve the adhesion of the coating [73]. A composite coating like Co–Ni–P/GO shows improved anticorrosion performance compared to the nonmodified Co-Ni-P coating. GO particles were embedded in the voids and crack sites of Co–Ni–P/GO, reducing the number of defects in the coating along with the adsorption of chloride ions and the corrosion rate of the substrate. Moreover, the insertion of GO distorts the channels, hindering penetration by corrosive media [52]. The next example is multilayer composite coatings such as SnZn–GO–SnZn and ZnNi–GO–ZnNi. The metallic coatings were electrodeposited, while the GO layers between them were deposited by drop casting. Three types of samples were fabricated by increasing the thickness of the lower metal layer and decreasing the thickness of the upper layer. Electrochemical measurements showed that the anticorrosion properties of all the composite coatings were enhanced compared to the pure metallic coatings. Moreover, corrosion current density was reduced as the thickness of the top metallic layer decreased. Thus, the measurements confirmed that the inertness and impermeability of GO effectively enhanced corrosion protection [57].

The anticorrosion abilities of organic and inorganic coatings were compared in different environments (seawater and crude oil-produced water). For this purpose, polymer composites of styrene-acrylonitrile, polyaniline, and few-layer graphene (SAN/PANI/FLG) and ZnO/GO were prepared as organic and inorganic coatings, respectively. Electrochemical analysis showed that in seawater, the organic and inorganic coatings reduced corrosion by up to 90% and 75% compared to the bare metal surface, respectively. In crude oil-produced water, corrosion was decreased by up to 95% and 10% by the organic and inorganic coatings, respectively. The better anticorrosion properties of SAN/PANI/FLG compared to ZnO/GO were attributed to the reduced pore formation and enhanced barrier properties [74]. Another example can be the modification of graphene with 3-aminopropyltriethoxysilane (APTES) which leads to the formation of covalent bonds between graphene and the copper substrate. The corrosion current density of these coatings decreased 20 times compared to uncoated Cu. The protective properties can be assigned to the impermeable nature of graphene [75].

## 4. Graphene as a Component of Organic Coatings

*4.1. Influence of Graphene Dispersion in the Polymer Resin, Coating Hydrophobicity, and Coating Adhesion Strength on the Anticorrosion Properties of Organic Coatings*

Graphene materials are often applied as nanofillers in organic coatings. One factor contributing to the enhanced anticorrosion properties of epoxy coatings containing GO is the good dispersion of GO flakes in the polymer matrix. The dispersion of GO flakes in the matrix results in tortuous diffusion pathways, which prevent aggressive electrolytes from reaching the substrate surface [76]. Pourhashem et al. investigated the effects of the coating preparation method and the amount of added GO on the dispersion of GO in the coating. The following two preparation methods were used: (I) GO was added to epoxy resin (R), and then hardener (H) was added [(R + GO) + H]; (II) GO was added to the hardener, and epoxy resin was then added [R + (H + GO)]. Due to the lower viscosity of the polyamide hardener compared to the epoxy matrix, the addition of GO flakes to the hardener (Method II) resulted in better dispersion than in Method I. Moreover, using a low amount of GO resulted in the best dispersion; high GO concentrations caused graphene aggregation, resulting in reduced corrosion resistance [77]. The improvement in anticorrosion properties caused by the uniform exfoliation of rGO in poly(methyl methacrylate) (PMMA) resin depends on the content of carboxylic groups in rGO. Studies have shown that the coatings with the highest contents of carboxylic groups are least susceptible to penetration by aggressive electrolytes [78].

The good dispersion of graphene materials in the polymer resin, the compact surface without defects, the superhydrophobicity, and the excellent adhesion to the polymer matrix significantly enhance corrosion protection. For example, the addition of GO to an epoxy coating was shown to reduce the crack length and cause the surface to become smoother, more uniform, and less wettable (the water contact angle increased from 68° to 75°) [79]. The thermal exfoliation of GO at 1000 °C for 30 s resulted in a substantial increase in the hydrophobicity (water contact angle of ca. 127°) of the graphene/epoxy composite, resulting in enhanced anticorrosion performance [80].

Traditional polymeric coatings are usually deposited from organic solution, which is harmful to people and the environment because significant amounts of volatile organic compounds are released. The use of water-based coatings could be a solution to this problem. However, despite their many advantages, these coatings are characterized by weak corrosion protection ability. The addition of nanofillers such as graphene can enhance the corrosion resistance of the water-based coatings. However, due to the van der Waals interactions and high specific surface area, the graphene is characterized by a tendency to aggregate. To avoid aggregation and obtain a uniform suspension, different chemical compounds can be used. Liu et al. obtained good graphene dispersion in a water-based epoxy coating thanks to the presence of sodium polyacrylate [81]. Well-dispersed graphene effectively limits the diffusion of aggressive ions. Moreover, the functionalization of graphene with titanate is an excellent way to obtain a uniform dispersion of graphene in a polyurethane (PU) composite. In this case, the addition of 0.2 wt.% functionalized graphene to PU improved the anticorrosion properties compared to the neat PU coating. When the graphene content was increased to 0.4 wt.%, the graphene sheets were self-aligned in the direction parallel to the substrate surface. This caused the full specific surface area of graphene to react with the electrolyte, preventing it from reaching the substrate surface [82].

GO has also been applied as a filler. The effects of GO flake size and oxidation level on the barrier properties of a water-based alkyd resin were examined. The researchers found that large flakes of GO with an appropriate oxidation degree could successfully enhance the protective performance of the coatings. Large GO flakes form a long pathway of permeability for the corrosive medium, while the high oxidation level of GO enables its good dispersion in the polymer resin while avoiding the formation of conductive paths [83]. The anticorrosion properties of waterborne epoxy (EP) coatings with four different graphene structures [graphene sheets without and with carbon black (Ga and Gb, respectively), GO powder, and graphene slurry] were compared. Among the samples, the EP-Ga

coating provided the best corrosion protection. Field-emission scanning electron microscopy analysis showed that the insertion of Ga in the resin resulted in a good distribution in the matrix along with a uniform surface with only minor cracks. Moreover, this coating was the least wettable, making it the least susceptible to penetration by water [84].

### 4.2. Positive and Negative Effects of Conductivity on Barrier Properties

Graphene is often incorporated in zinc-rich coatings (ZRCs; Figure 3). The protective properties result from two protection mechanisms. The first mechanism is based on cathodic protection since zinc particles are in electrical contact with each other and the surface. Because the zinc particles have a more negative potential than the substrate, they undergo oxidation and prevent the dissolution of the substrate. The second mechanism is based on the mechanical barrier provided by the impermeable layer of insoluble zinc products that forms on the surface. High zinc content (>80%) is necessary to ensure good electrical contact. However, this also changes the coating properties, including reducing coating flexibility, reducing substrate adhesion, and increasing viscosity. If oxidation results in an excessive amount of zinc corrosion products, the conductivity between zinc particles and the surface is lost, and the cathodic protection disappears. Therefore, some amount of filler can be used to substitute for zinc particles [85]. Numerous studies have confirmed the enhancement in the anticorrosion properties of ZRC using graphene. The addition of graphene reinforces the electrical connection, extending the duration of cathodic protection [86]. Moreover, the role of graphene in zinc-rich epoxy coatings depends on the amount of zinc and graphene added [87,88]. The addition of a low amount of Zn (10 wt.%) makes the mechanism based on the sacrificial dissolution of Zn ineffective. When the content is increased to 40 wt.%, good anticorrosion properties are obtained [87]. Among four ZRC samples containing different amounts of graphene (0.25, 0.5, 0.75, and 1 wt.%), the best anticorrosion properties were obtained for the sample with 0.75 wt.% graphene, which can be attributed to the improved electrical connection between adjacent zinc particles [88]. The corrosion resistance of ZRC with GO or rGO was also investigated. Electrochemical tests indicated that the protective properties of these coatings were improved with respect to ZRC; however, rGO in ZRC was characterized by the highest corrosion resistance. This result can be attributed to the substantial enhancement in electrical contact between particles [89].

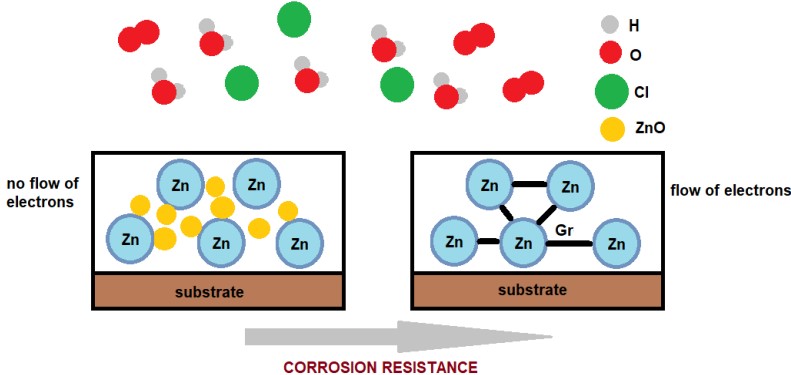

**Figure 3.** Mechanism protection of zinc-rich polymer coatings.

In the case of not-zinc-rich polymer coatings, the introduction of compounds with increased conductivity adversely affects the protective properties. Namely, electrochemical tests showed that a rGO/EP coating was characterized by short-term corrosion protection, whereas the GO/EP coating provided long-term protection [90].

## 5. Modification of Graphene-Based Polymer Coatings

*5.1. Effects of the Dispersibility of Organic-Functionalized Graphene, Coating Hydrophobicity, and Coating Adhesion Strength on the Protective Properties*

Coupling GO with silane is one way to modify graphene for use in protective coatings. Pourhashem et al. studied the anticorrosion performance of a $SiO_2$–GO hybrid in an epoxide matrix. A high contact angle [91,92] and high adhesion strength resulting from strong Si–O bonds between the epoxy coatings and metal substrate [91] are important for corrosion resistance. The corrosion protection mechanism of these coatings is also based on the formation of tortuous pathways (Figure 4) [91,92]. In addition, the method used to prepare the $SiO_2$–GO hybrid affects the final protective properties. Ramezanzadeh et al. synthesized nanohybrids using two methods: (I) the insertion of GO flakes into silane solution followed by silanization for 24, 48, and 72 h; and (II) the incorporation of GO flakes after silanization for 24, 48, and 72 h. The results showed that Method I resulted in better anticorrosion properties. The silica molecules acted as spacers between the GO flakes and enhanced the dispersion of GO in the polymer matrix. For both preparation methods, the best anticorrosion performance was observed after silanization for 48 h [93]. GO in polymer coatings has also been modified by aminosilane. The functionalized coatings exhibited markedly higher anticorrosion performance compared to the unfunctionalized coatings, and different mechanisms of corrosion protection were considered: (1) the combination of aminosilane with GO provides strong bonds with the epoxy resin, leading to an increase in cross-link density; (2) the higher adhesion properties compared to GO/EP and blank EP, which can be attributed to the presence of Si–O–C bonds between the coating and metal surface; and (3) the improved dispersion resulting from the introduction of A-GO in the organic matrix, although it is limited by the amount applied a nanofiller [94]. The barrier properties of the coatings also depend on the type of silane precursor. Pourhashem et al. investigated the effects of using APTES and (3-glysidyloxypropyl) trimethoxysilane (GPTMS) as silane precursors on the protective properties of organic coatings. Regardless of the type of silane coupling agent, the resulting coatings exhibited excellent anticorrosion performance compared to the neat EP and EP/GO coatings. The incorporation of these nanofillers in the polymer matrix contributes to increasing the cross-link density of the layer along with the contact angle and adhesion strength. However, GO modified with APTES showed the best barrier properties among the tested silane precursors. The amine groups of APTES form hydrogen bonds with the glycidyl groups in the epoxy resin, resulting in a dense polymer network [95].

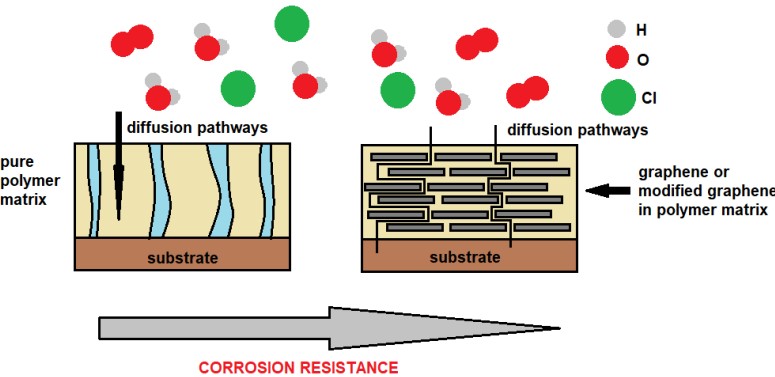

**Figure 4.** Mechanism protection of graphene polymer-based coatings.

Silane-functionalized GO has been applied in epoxide coatings as well as in polyurethane and acrylic matrices. The hydrophobic nature of silane-functionalized GO along with its resistance to delamination contributes to improving the corrosion resistance of polyurethane coatings [96]. Inserting the correct amount of GO functionalized with organosilanes in acrylic coatings in a cataphoretic bath resulted in coatings with negligible defects. Furthermore, incorporating the optimized content of

silane-functionalized GO improved the protective performance, effectively blocking the penetration of corrosive media [97].

In addition to GO nanocomposites with silanes in epoxy coatings, many studies have explored GO composites with nitrogen compounds. For example, the corrosion protection properties of epoxy coatings were reinforced with GO–poly(urea–formaldehyde) (GUF) composites [98]. The introduction of urea–formaldehyde caused the hydrophilic GO to become lipophilic, which enhanced the exfoliation and dispersion of the graphene flakes while reducing aggregation. Polyaniline (PANI)–GO nanocomposites have also been applied in epoxy coatings. Researchers tested the behaviors of different nanocomposites containing various amounts of GO (3, 6, 12, and 24 wt.%) in a corrosive medium. Among the samples, the highest corrosion resistance was achieved by the coating containing 12 wt.% GO [99], which was attributed to the good dispersion in that sample. These types of coatings are protected from aggressive chloride ions (testing was conducted in NaCl) by the negative surface charge of PANI–GO in the epoxy matrix. Combinations of p-phenylenediamine with graphene flakes of various sizes in polymer resins were also studied. In general, all of the coatings exhibited improved anticorrosion performance compared to neat epoxy coating, although the best properties were obtained by the coatings with medium-sized graphene flakes [100].

Functionalized graphene oxide was also used in polystyrene (PS) and polycaprolactone (PCL) matrix. Modification of GO with polyvinyl caused improvement in corrosion resistance compared to bare polystyrene coatings. Corrosion protection efficiency was increased from 37.90% for pure PS to 99.53% for polystyrene coating with 2% content of functionalized GO. The protective properties can be assigned to the prolonged pathways for gas diffusion which formed as a result of the complete exfoliation of modified GO [101]. In turn, 8-bromo-[1,2,4]triazolo[1,5-a]pyridin-2-amine (BTP) was used for the functionalization of GO in PCL coatings. Superior corrosion resistance resulted from good dispersion and exfoliation in polymer resin. Moreover, these coatings were more hydrophobic and adhesive to the substrate surface [102].

The literature provides a lot of information about the modification of GO in waterborne coatings. The effects of GO functionalized with lysine [103], polydopamine [104,105], polyetheramine [106], isophorne diisocyante [107], and phytic acid [108] on coating barrier properties have been studied. The improved corrosion resistance of the polymer coating after the introduction of graphene modified with lysine, polyetheramine, isophorone diisocyanate, or phytic acid can be attributed to the more cross-linked layer and blocking of diffusion pathways. Modifying GO with polydopamine can improve the protective properties in comparison to nonfunctionalized coatings. This effect is associated with the well-dispersed particles in the polymer resin, which result in the formation of tortuous pathways [105]. The corrosion resistance also stems from the reduction in wettability. For polydopamine-GO in waterborne polyurethane coatings, the anticorrosion properties result from both the decreased hydrophilicity and the improved adhesion [104].

## 5.2. Effects of the Dispersibility of Inorganic Functionalized Graphene, Coating Hydrophobicity, and Coating Adhesion Strength on the Protective Properties

In organic matrices, GO can be decorated by inorganic compounds such as titanium dioxide [109], neodymium [110], fluorine [111], zirconia dioxide [112], $Fe_3O_4$ [113], alumina [114,115], cerium oxide [116,117], hexagonal boron nitride (hBN) [118], zinc phosphate (ZP) [119], and $Si_3N_4$ [120]. Due to the good dispersion of $TiO_2$–GO, $ZrO_2$–GO, and $Al_2O_3$–GO in the polymer matrix as well as the sheet structures of the composites, these coatings show enhanced anticorrosion performance compared to GO/EP or bare epoxy coatings. The synthesis of GO with cerium oxide and nitrogen-containing compounds is an excellent way to improve the anticorrosion performance. As expected, combining $CeO_2$ with PANI leads to the formation of zigzag pathways while also hindering the free access of aggressive electrolyte to the substrate surface [116]. $CeO_2$ was also mixed with APTES, obtained nanocomposite in epoxy coating decreased wettability properties, and increased adhesive properties, simultaneously enhancing protective performance [117]. The comodification of GO/$Fe_3O_4$ nanocomposites with

dopamine (DA) and silane (KH550) resulted in excellent anticorrosion properties. The functional groups formed as a result of the comodification (e.g., $NH_2$ and OH) improved dispersion in the polymer matrix and enhanced the interfacial adhesion strength between the nanoparticles and epoxy resin. As a result of the crosslinking reaction between GO-$Fe_3O_4$@poly (KH550 + DA) hybrid and the epoxy, the coatings became less hydrophilic [113]. The introduction of $Si_3N_4$ in waterborne polymer resins both improves dispersion and decreases wettability [120]. The use of fluorographene (FG) in epoxy coatings results in a superhydrophobic surface (water contact angle of ~154°) that repels water molecules and thus results in enhanced anticorrosion performance [111].

The hydrophobicity of waterborne epoxy coating modified GO with hexagonal boron nitride makes these coatings resistant to penetration of aggressive ions [118]. Nd–GO provides a good barrier layer in epoxy coatings, preventing the aggressive electrolytes from reaching the surface; this indicates the strong physisorption of the GO + Nd-based coating on the alloy surface [110]. The anticorrosion properties of zinc phosphate modified with GO in water-based polyurethane coatings can be attributed to the extra barrier in the film [119].

### 5.3. Positive and Negative Aspects of Enhanced Conductivity

The addition of PANI-grafted graphene to ZRCs can effectively enhance the barrier properties due to the synergistic effect of cathodic and anodic protection. The deposition of a conductive polymer on the GO surface improves the barrier properties and enhances electrical contact between the molecules. The latter effect prolongs the active time of the sacrificial zinc mechanism [121,122].

The electrical properties of the dopants applied in water-based polyurethane resin are crucial for the protective performance of the coating. Boron-doped graphene materials exhibit remarkably better long-term corrosion protection compared to nitrogen-doped graphene composites. This can be attributed to the insulation properties of boron, which inhibit galvanic reactions. In turn, doping with nitrogen enhances the conductivity of the graphene coating, which can accelerate corrosion [123]. Reduced sulfonated graphene has also been applied in waterborne polyurethane coatings. Its introduction reduced the conductivity of the coating, preventing electrons from penetrating the coating [124]. The functionalization of GO with metronidazole (MET), a corrosion inhibitor, substantially enhanced the protective properties of the coating [125] by reducing its conductivity, thereby prolonging the lifetime of the surface.

In Tables 4 and 5, a comparison of corrosion resistance between coatings as well as a comparison of their long-term anticorrosion performance is demonstrated, respectively. Additionally, in Figure 5, the direction of the development of anticorrosion protective coatings is presented.

**Table 4.** Comparison of corrosion resistance between coatings.

| Coatings | Adhesion Strength | Wettability | Electrochemical Impedance Spectroscopy | Potentiodynamic Polarization | Literature References |
|---|---|---|---|---|---|
| N-doped GO (N–GO/SS) | Not available | 42.5° (GO), 121.5° (NGO) | Charge transfer resistance ($R_{CT}$) ($\Omega \cdot cm^{-2}$) 5930 (SS), $2.138 \times 10^4$ (GO/SS), $2.1834 \times 10^5$ (N–GO/SS) | $I_{corr}$ ($\mu A \cdot cm^{-2}$): 5.660 (SS), 0.951 (GO/SS), 0.007 (N–GO/SS) | [69] |
| Silane coupling (EP/SiO$_2$–GO) | 8.5 ± 0.5 MPa (EP), 11.1 ± 1 MPa (EP/GO), 17.7 ± 1.5 MPa (EP/GO–SiO$_2$) | 73.1 ± 2° (EP), 70.7 ± 2° (EP/GO), 84.3 ± 1° (EP/GO–SiO$_2$) | $R_{CT}$ for EP/GO–SiO$_2$ higher than other coatings, | $I_{corr}$ ($\mu A \cdot cm^{-2}$) 14.6 (EP), 0.22 (EP/GO), 0.09 (EP/GO–SiO$_2$) | [91] |
| Aminosilane coupling (EP/A-GO) | 8.5 ± 0.5 MPa (EP), 11 ± 1 MPa (EP/0.1GO), 17.7 ± 0.5 MPa (EP/0.1A-GO) | Not available | $R_{CT}$ for EP/0.1A-GO higher than other coatings, | Not available | [94] |
| Nitrogen coupling (GUF)/EP | Not available | Not available | $R_{CT}$ ($\Omega \cdot cm^{-2}$): $1.85 \times 10^{10}$ (EP), $6.22 \times 10^{10}$ (EP/GO), $7.2 \times 10^{11}$ (EP/GO-GUF) | Not available | [98] |
| Nitrogen coupling (EP/FGO) | Enhancement of adhesion strength by incorporation functionalized graphene oxide | 91.7° (PCL), 101.7° (PCL/GO), 106.0° (PCL/FGO) | $R_{CT}$ ($\Omega \cdot cm^{-2}$) $1.942 \times 10^3$ (PCL), $3.946 \times 10^7$ (PCL/GO), $1.784 \times 10^8$ (PCL/FGO) | $I_{corr}$ ($\mu A \cdot cm^{-2}$): 5.138 (PCL), 0.2132 (PCL/GO), 0.01033 (PCL/FGO) | [102] |
| With zirconia dioxide (EP/GO–ZrO$_2$) | 10.73 MPa (EP), 11.13 MPa (EP/GO), 12.52 MPa (EP/GO–ZrO$_2$) | Not available | $R_C$ (coating resistance) higher for composite coatings than pure EP | $I_{corr}$ ($\mu A \cdot cm^{-2}$) 3.54 (EP), 0.49 (EP/GO), 0.37 (EP/GO–ZrO$_2$) | [112] |
| With fluorographene (EP/FG) | Enhancement of adhesion strength by incorporation of FG | 82° (EP), 116° (EP/GO), 154° (EP/FG) | $R_{CT}$ ($\Omega \cdot cm^{-2}$) $1.38 \times 10^5$ (EP) $2.5 \times 10^6$ (EP/GO) $5.24 \times 10^8$ (EP/FG) | $I_{corr}$ ($\mu A \cdot cm^{-2}$) 1.05 (EP) 0.144 (EP/GO) 0.000501 (FG/GO) | [111] |
| With hexagonal boron nitride (EP/GO–hBN) | Not available | 86.9° (waterborne epoxy coating; WBE), 92.8° (WBE/GO), 94.9 ÷ 98.0° (WBE/GO–hBN, depend on added amounts of GO) | $R_C$ ($\Omega \cdot cm^{-2}$): $4.95 \times 10^5$ (WBE), $2.89 \times 10^6$ (WBE/GO), $4.05 \times 10^6$–$2.17 \times 10^6$ (WBE/GO–hBN, depend on added amounts of GO) | Not available | [118] |
| With zinc phosphate (waterborne polyurethane (WPU)/GO-ZP) | Not available | Not available | $R_{CT}$ ($\Omega \cdot cm^{-2}$): 826 (WPU), 3762 (WPU/ZP), 4726-11040 (WPU/GO-ZP, depend on added amounts of GO) | $I_{corr}$ ($\mu A \cdot cm^{-2}$) 49.4 (steel), 24.2 (WPU), 5.54 (WPU/ZP), 2.73 ÷ 0.441 (WPU/GO-ZP, depend on added amounts of GO) | [119] |

**Table 5.** Comparison of long-term anticorrosion properties between different coatings.

| Coatings | Description | Literature References |
|---|---|---|
| N-doped graphene coating (NG) | Graphene (PG) and three samples NG (NG1, NG2, NG3, the different doping concentration of nitrogen, the flow rate of $NH_3$ 1, 2, 4 sccm, respectively) deposited by CVD. Exposure time in the air: 2 weeks, 1 month, 3 months | After two weeks, no obvious difference in the surface morphology for PG, whereas for NG1 and NG3—inhomogeneous corrosion on the form of patches, for NG2 no signs of corrosion. After 1 month: corroded area for PG, 50%; NG1, 10%; NG2, 30%; NG2, no color change of surface. After 3 months: PG—severely corroded, NG coatings less corroded than PG. | [69] |
| APTES/Gr | Samples: APTES and APTES/Gr coatings (with different content of graphene: 0.1, 0.5, 1, 5 wt.% content). Exposure time in 3.5% NaCl up 480 h | Water permeation and breakpoint frequencies are used to evaluate the electrochemical activity of the surface. The uptake of water decreases with the increase of graphene content at a given time. Breakpoint frequencies occur immediately after immersion for APTES coatings, whereas for graphene-based coating gets delayed as the content of graphene increases. For APTES/Gr (5%), breakpoint frequencies occur after 48 h exposure, making this coating the most resistant to corrosion. | [126] |
| $ZrO_2$–GO/ZnAl coatings | | After 480 h salt spray test: <br> - For ZnAl coatings, visible traces of red rust; <br> - For $ZrO_2$/ZnAl—small amount of red rust; <br> - For GO–$ZrO_2$/ZnAl—no obvious red rust. | [73] |
| GO/EP | Cotaings of three samples with different amounts of GO: 0.125, 0.25, 0.5 %. Exposure time in 3.5% NaCl: 1, 34, 64 days. | $R_c$ ($\Omega \cdot cm^{-2}$) for coatings after 1 days' immersion in chloride solution: <br> EP—$5.387 \times 10^7$, EP/0.125GO—$4.613 \times 10^7$, EP/0.25GO–$1.771 \times 10^8$, EP/0.5GO—$8.639 \times 10^7$. <br> $R_c$ ($\Omega \cdot cm^{-2}$) After 34 days: <br> EP—$6.72 \times 10^6$, EP/0.125GO—$2.797 \times 10^6$, EP/0.25—$6.722 \times 10^9$, EP/0.5GO—$1.127 \times 10^7$ <br> $R_c$ ($\Omega \cdot cm^{-2}$) After 64 days: <br> EP—$2.757 \times 10^5$, EP/0.125GO—$2.094 \times 10^6$, EP/0.25—$2.257 \times 10^7$, EP/0.5GO—$2.037 \times 10^7$ | [76] |
| GO and rGO in EP coating | Time of exposure in 3.5% NaCl: 1 h, 24 h | $R_{CT}$ ($\Omega \cdot cm^{-2}$): <br> rGO/EP 1 h: $1.455 \times 10^4$ <br> rGO/EP 24 h: 1450 <br> GO/EP 1 h: 2470 <br> GO/EP 24 h: 2010 | [90] |

<div align="center">

**Table 5.** *Cont.*

</div>

| Coatings | Description | Literature References |
|---|---|---|
| Graphene (Gr) in zinc-rich epoxy (ZRC) coating | Two kinds of coatings with different time of exposure (1, 10, 25 days) with the same amount of graphene (wt. 0.6%) | $Rc\ (\Omega \cdot cm^{-2})$: <br> ZRC1: 6630 <br> ZRC10: 3350 <br> ZRC25: 3030 <br> Gr0.6/ZRC1: 6630 <br> Gr0.6/ZRC10: 6100 <br> Gr0.6/ZRC25: 11,200 | [86] |
| Multilayer graphene (FMLG) in ZRC | Five kinds of coatings: pure ZRC and four coatings with different amounts (0.25, 0.5, 0.75, 1.0 wt.%) of graphene in ZRP | Salt spray test after 500 h: <br> Pure ZRC: red rusts appeared at scratched regions, some blisters on the surface, <br> FMLG0.25/ZRC: red rust at the scratched region, no blisters on the surface <br> FMLG0.5/ZRC: small amount of red rust on the surface <br> FMLG0.75/ZRC: no red rust on the surface. <br> FMLG1.0/ZRC: some blisters on the surface | [88] |
| PANI/EP | Six kinds of coatings: pure EP, PANI/EP, and four coatings with different amounts of added GO (3, 6, 12, 24 wt.%). Time of exposure in 3.5% NaCl: 2, 24, 144, 194 h | $Rc\ (\Omega \cdot cm^{-2})$ <br> EP 2 h $1.01 \times 10^4$ <br> EP 24 h $2.13 \times 10^3$ <br> EP 144 $9.90 \times 10^3$ <br> EP 192 h $1.00 \times 10^4$ <br><br> EP/PANI 2 h $9.29 \times 10^3$ <br> EP/PANI 24 h $1.67 \times 10^4$ <br> EP/PANI 144 $3.49 \times 10^4$ <br> EPPANI 192 h $6.82 \times 10^3$ <br><br> EP/PANI-GO (3 wt.%) 2 h $1.45 \times 10^5$ <br> EP/PANI-GO (3 wt.%) 24 h $8.01 \times 10^3$ <br> EP/PANI-GO (3 wt.%) 144 $2.73 \times 10^5$ <br> EPPANI-GO (3 wt.%) 192 h $3.41 \times 10^5$ <br><br> EP/PANI-GO (6 wt.%) 2 h $2.67 \times 10^7$ <br> EP/PANI-GO (6 wt.%) 24 h $2.93 \times 10^6$ <br> EP/PANI-GO (6 wt.%) 144 $6.86 \times 10^5$ <br> EPPANI-GO (6 wt.%) 192 h $4.37 \times 10^5$ <br><br> EP/PANI-GO (12 wt.%) 2 h $2.40 \times 10^7$ <br> EP/PANI-GO (12 wt.%) 24 h $2.70 \times 10^7$ <br> EP/PANI-GO (12 wt.%) 144 $4.91 \times 10^6$ <br> EPPANI-GO (12 wt.%) 192 h $2.70 \times 10^6$ <br><br> EP/PANI-GO (24 wt.%) 2 h $2.78 \times 10^5$ <br> EP/PANI-GO (24 wt.%) 24 h $5.73 \times 10^4$ <br> EP/PANI-GO (24 wt.%) 144 $6.36 \times 10^4$ <br> EPPANI-GO (24 wt.%) 192 h $3.21 \times 10^5$ | [99] |

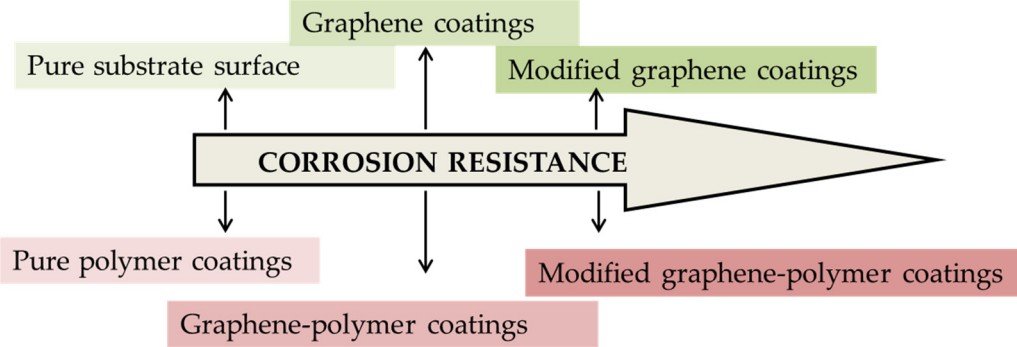

**Figure 5.** The direction of development of anticorrosive protective coatings.

## 6. Protective Mechanism of Graphene Coatings, Problems Resulting from Coating Preparation, and the Effects of These Problems on Corrosion Resistance

Numerous studies have shown that graphene coatings effectively protect the substrate from corrosion. The protective mechanism of the graphene coating is based on the physical barrier provided by graphene, which prevents aggressive ions from reaching the surface of the substrate. However, obtaining perfect graphene layers without defects is particularly difficult. Grain boundaries and edges serve as sites for electrolyte penetration. The presence of defects, cracks, and/or wrinkles formed during graphene deposition also markedly decreases corrosion resistance [127]. Moreover, studies have shown that defects can promote short-term protective performance; however, their presence weakens long-term corrosion protection. In addition, the high conductivity of graphene coatings can accelerate corrosion [128]. Galvanic microcells are formed, in which exposed metal acts as an anode. However, the defect density and protective properties of CVD-based graphene coatings can be modulated by controlling the cooling rate and flow of hydrogen [129]. The use of a low cooling rate, regardless of the flow of hydrogen, can inhibit the formation of graphene layers. In contrast, the application of a high cooling rate and no hydrogen flow can reduce the formation of wrinkles and defects simultaneously enhancing the protective properties. Further, the formation of multilayer graphene has been shown to compensate for the defects in monolayer graphene films.

The anticorrosion mechanism of electrophoretically deposited GO coatings is also based on the barrier that separates the coated material from the environment. As mentioned above, the corrosion resistance of electrodeposited GO coatings is determined by the chemical composition of the coating and the parameters of the coating process. The process parameters, especially the suspension pH, can change the deposition mechanism. Under basic conditions, the GO sheets are parallel to the metal substrate, forming a rug-like morphology. In contrast, under acidic conditions, the GO sheets from face-to-face restacked multilayers with a brick-morphology [130]. Independent of the morphology, twisted pathways for aggressive electrolyte are formed, thereby reducing corrosion.

A completely different type of anticorrosion film is polymer coatings containing graphene or functionalized graphene. The protective effect of these coatings is based on the blocking of pores formed during the curing process by added nanofillers. The good distribution of graphene nanofillers plays a crucial role in the formation of tortuous pathways, which prevent penetration by the corrosive medium. The protective property depends on the amount of added graphene material; an excessive quantity of added graphene can lead to the aggregation of graphene sheets in the polymer resin.

The protective properties of ZRCs are associated with the sacrificial dissolution of zinc, which depends on the electrical contact between zinc molecules/particles and the substrate surface. The introduction of graphene or modified graphene between zinc particles provides and prolongs the electrical contact, thereby enhancing cathodic sacrificial protection. In turn, the introduction of nanofillers to not-zinc-rich polymer resins (e.g., reduced GO or nitrogen-doped GO) increases the conductivity and therefore accelerates corrosion.

## 7. Conclusions

This review presents the current state-of-the-art in graphene coatings, graphene-based organic coatings, the modification of graphene coatings, and the effect of graphene functionalization on the anticorrosion properties. Because standalone graphene coatings as well as graphene in polymer coatings show protective properties, however, these coatings suffer from the presence of pores and cracks and poor dispersibility graphene in polymer resins respectively, resulting in reduced protective performance. The functionalization of graphene oxide with different chemical organic or inorganic compounds significantly improved the anticorrosion properties. The mechanism activity of formed coatings consists of the enhancement of the hydrophobicity and adhesion strength of the protective coating as a result of the incorporation of modified graphene. Moreover, in the case of the polymer coatings, the increased corrosion resistance results also from an improvement of the dispersibility of graphene. This leads to the formation of zigzag pathways, hindering the transport of aggressive electrolyte to the substrate surface.

All earlier described studies demonstrate the increased protective properties, which they result from the aforementioned factors. However, there are only a few reports discussing the enhancement of anticorrosion performance caused by all factors. Therefore, the future directions of research should focus on the search of such chemical compounds enhancing all requirements of the perfect protective layer. Furthermore, the presented coatings in this review are not appropriate for large scale production due to the high costs, time-consuming process as well as applying toxic and harmful reagents. Therefore, future studies should also focus on the enhancement of corrosion resistance, but using more environmentally friendly reagents and more economical and easier routes of production.

**Author Contributions:** Conceptualization, K.O.; writing—original draft preparation, K.O.; writing—review and editing, M.L.; supervision, M.L. All authors have read and agreed to the published version of the manuscript.

**Funding:** This research received no external funding.

**Conflicts of Interest:** The authors declare no conflict of interest.

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
