# Peer review of "Review of the Application of Graphene-Based Coatings as Anticorrosion Layers"

_coatings, doi:10.3390/coatings10090883_

Round 1

Reviewer 1 Report

In the submitted manuscript, the authors provide a review on the application of graphene-based coatings as anti-corrosion layers.

In general, the scalable formation of highly reduced GO and graphene films is crucial to the development of graphene-based devices and such analyses will significantly push forward the current status-quo. The topic studied results discussed, and the quality of the experiments is good, however, it lacks in-depth representation for the area of graphene-based materials applications. My main concerns regarding the manuscript are mentioned below:

  1. The introduction is lacking in certain aspects. The discussion primarily focusses on equilibrium based processing techniques. The primary issues with these techniques – namely generation of wrinkles and exposure of the whole wafer towards reactive chemicals needs to be mentioned. In retrospect, the nonequilibrium methods of graphene oxide fabrication need to be discussed. In general, laser annealing leads to the formation of wrinkle-less graphene oxide films (ACS Appl. Mater. Interfaces 2019, 11, 27, 24318-24330). There have been reports produced for reducing the GO itself using laser annealing. (Carbon, 52, pp.574-582) With this method it is possible to achieve electron mobility and oxygen content modulation by changing the irradiation energy density. (Carbon, Volume 170, 2020, pp. 327-337)
  2. The authors need to present the structural information of these films produced by varied techniques mentioned. This can be presented as a sampled Raman spectroscopic investigations and calculation of ID/IG ratio.
  3. The structural analysis will provide information regarding defect concentration and oxygen content in these films. This information then needs to be correlated with anti-corrosion characteristics.
  4. In corrosion analysis some sort of quantification needs to be provided to compare across results. Right now the report is mostly textual lacking any real analysis. The authors can either find results from reports with thickness reduction over time, pit diameter increase over time or the standardized Electrochemical Impedance Spectroscopy data.

These concerns need to be addressed before publication in an archival journal like MDPI Coatings.

Reviewer 2 Report

The review by Ollik and Lieder has some merit but there are recent reviews on the topic. The topic fits well the scope of the journal. There are several works that are missed from the review. The critical aspects and the discussions on the future directions need to be strengthened. Overall, the review could be publishable in Coatings but there are many minor and major issues to be addressed.

1, The review needs to have a critical edge. The drawbacks and shortcomings of the graphene-based anti-corrosion layers should be mentioned. In particular, it may accelerate dangerous localized corrosion that can seriously weaken the coated metals (10.1038/nnano.2017.187).

2, The authors should discuss the long-term performance of these coatings, and gather all studies in a single table to compare the results include also 10.1016/j.flatc.2016.08.003.

3, Examples to make graphene coatings more robust should be also mentioned (10.1016/j.matt.2019.04.005; 10.1021/acsnano.8b02373).

4, Some anticorrosive coatings, e.g. on copper was missed and should be included in the review (10.1039/C7RA10167H)

5, A comprehensive review on graphene-based anti-corrosive coatings was recently published (10.1016/j.cej.2019.05.034). The authors need to mention this and highlight what the current review adds to the discussion, and in what way it is different and more than what was published before.

6, Table 1 should include in separate columns both the advantages and disadvantages of each method presented in the table itself. This will assist the readers to quickly grasp the pros and cons of the different methodologies and ultimately assist in decision making.

7, Figure 1 is incomplete and better to have a short paragraph on applications rather than showing the figure, which can be deleted. See next comment.

8, The manuscript starts in medias res. A general but brief introduction to graphene should be given, and its widespread applications should be exemplified: electronics 10.1021/acsapm.0c00539; breaking emulsions 10.1016/j.memsci.2020.118007; gas separation 10.1021/acsapm.9b00426; catalysis 10.1039/D0GC01274B; electrodialysis 10.1039/C8TA09160A; biomimetics 10.1039/C9GC02266J; nanofiltration 10.1021/acsami.8b03591.

9, Nanocomposites with excellent anti-corrosion properties should also be briefly mentioned (10.1039/c3py00825h, j.matchemphys.2019.122050).

10, The different application areas of the anti-corrosion coatings should be shown in a figure and their individual requirements should be listed.

11, The adhesion strength of the graphene coatings should be discussed in more depth, and the parameters affecting it should be analysed and suggestions made for improvements. In general, future directions are expected in a review, and the authors failed to show the weaknesses of the coatings and where the field should be heading.

12, The conclusion section is somewhat vague. Include the main achievements and pioneering research as well as mention future directions more tangibly.

Reviewer 3 Report

I have read with initial interest the manuscript from Ollik et al, since I believe the topic is of graphene-based coatings is absolutely of great interest. Albeit the efforts from the authors, the manuscript is far for being acceptable for publication, as I would recommend the authors need to carry out the following changes:

  • Cite the review from Cui Chemical Engineering Journal 373, 2019, 104.
  • Consider some more cautious reports about graphene-coating, Nature Nanotechnology 12, 2017, 834.
  • Prepare new versions of Figure 1 and Table 1, as their current form are both of low quality (especially Figure 1).
  • Insert some exemplary figures in each section of the review. They are need in order to make the review more interesting to read.

Round 2

Reviewer 1 Report

The authors have taken all of my points into consideration and have done a good job at making the manuscript more data driven and cohesive towards graphene-based coating applications towards corrosion resistant materials. 

Reviewer 2 Report

The authors have done a thorough revision and addressed the comments.

Reviewer 3 Report

The authors have consistently improved the quality of the manuscript by following the requested changes. I believe that the manuscript is now of sufficient quality to be published in the current form.